# Effects of dual-task interference on swallowing in healthy aging adults

Rahul Krishnamurthy[1,2], Rhea Philip[3], Radish Kumar Balasubramanium[1,2], Balaji Rangarathnam[4] *

**1** Department of Audiology and Speech-Language Pathology, Kasturba Medical College, Mangalore, India, **2** Manipal Academy of Higher Education, Manipal, Karnataka, India, **3** Springfield Rehabilitation, Cochin, India, **4** Department of Speech-Language Pathology, Midwestern University, Downers Grove, Illinois, United States of America

\* branga@midwestern.edu

**Data Availability Statement:** All relevant data are within the paper and its Supporting Information files.

**Funding:** The authors received no specific funding for this work.

## Abstract

A wide body of literature has demonstrated that the neural representation of healthy swallowing is mostly bilateral, with one hemisphere dominant over the other. While several studies have demonstrated the presence of laterality for swallowing related functions among young adults, the data on older adults are still growing. The purpose of this paper is to investigate potential changes in hemispheric dominance in healthy aging adults for swallowing related tasks using a behavioral dual-task paradigm. A modified dual-task paradigm was designed to investigate the potential reduction in hemispherical specialization for swallowing function. Eighty healthy right-handed participants in the study were divided into two groups [Group 1: young adults (18–40 years) and Group 2: older adults (65 and above)]. All the participants performed a timed water swallow test at baseline and with two interference conditions (silent word repetition, and facial recognition). The results of the study revealed the following 1) a statistically significant effect of age on swallow performance; 2) statistically significant effect of each of the interference tasks on two of the swallow measures (VPS and VPT) in younger adults; and 3) no significant effect of the interference tasks on the swallowing performance of older adults. These findings suggest that aging substantially affects swallowing in older individuals, and this potentially accompanies a reduction in the hemispheric specialization for swallowing related tasks.

## Introduction

Research on neurological control of swallowing implicates numerous regions in the brain. These include, but are not limited to, the precentral gyrus, postcentral gyrus, premotor area, supplemental motor area, anterior cingulate cortex, operculum, insula, precuneus, cuneus, prefrontal area, temporal cortex, cerebellum, brainstem, frontal cortex, internal capsule, association areas, thalamus, and the basal ganglia [1–10]. In addition to the involvement of these neural structures and pathways, hemispheric dominance is a feature of swallowing, as in many other physiological functions [11,12].

**Competing interests:** The authors have declared that no competing interests exist.

## Hemispherical specialization for swallowing

Functional imaging and behavioral methods have been utilized to understand neural mapping/hemispheric specialization for swallowing. Research utilizing functional imaging methods such as functional magnetic resonance imaging [fMRI] [1,2,13–15], Positron emission tomography [PET] [3], magnetoencephalography [MEG] [16] has informed that the neural representation of swallowing is largely asymmetric and bilateral. That is, swallowing may not be localized to one specific hemisphere (right or left) across subjects, but within each subject one hemisphere appears to be more dominant than the other [2–5,11]. These findings are also supported by clinical reports which have demonstrated that unilateral ischemic strokes of either cerebral hemisphere can result in distinct patterns of swallowing impairments [17–20]. Specifically, left hemisphere damage may be associated with an impairment of the oral stage of swallowing, whereas right hemisphere impairments may be associated with pharyngeal stage dysmotility, aspiration, and persistent dysphagia [21]. Several other studies [22–26] report no such hemispheric differences in swallowing behavior due to brain injury.

Behaviorally, dual-task paradigms have been used to study lateralization for swallowing [27–29]. The dual-task paradigm is a neuropsychological method that indirectly investigates lateralized cortical systems by comparing baseline task performance with competing/interference conditions. Cerebral underpinnings of a dual-task interference have been explained using several theories. Among existing theories, the functional cerebral space model [30] states that the coactivation of functionally overlapping neural substrates will result in a performance decline of one of the two concurrent activities. Others have speculated that "hemispheric overload" [31] or "competition of resources" [32] results in the decrement of response. That is, if two tasks are sharing resources within the same hemisphere, there would be a compromised allocation of attentional or processing resources. Due to the limited functional capabilities of the neural system, there would be a decrement in at least one concurrent behavioral response from baseline performance.

Using the dual-task paradigm, Daniels et al. [27] offer partial support for the bilateral representation of swallowing in younger adults. Daniels et al. [28] further expanded on their earlier study to report that the hemispheric dominance for swallowing might be inconsistently lateralized, with the two hemispheres playing different roles in the control of swallowing. The authors reported a significant reduction in the measure of volume per swallow during a silent word repetition (left hemisphere activating task), and a significant reduction number of swallows during a line orientation task (right hemisphere activating task), thereby proposing that the left hemisphere regulates volume-related parameters of swallowing and that the right hemisphere regulates time-related parameters. Similar findings have also been reported by Balasubramanium et al. [29] in younger adults.

## Aging effects on swallowing and swallowing related laterality

Extant research [33–35] suggests that the swallowing function declines with age and a majority of age-related changes in swallowing occur after the age of 60 years [36]. Understanding age-related change in swallowing is important for several reasons. Firstly, there is a global rise in the population of older adults. In 2019, it was estimated that one in 11 people would be aged 60 years or above in the world. Currently, it is projected that the number of older adults in the world would be 1.4 billion in 2030 and 2.1 billion by 2050 and could rise to 3.1 billion in 2100 [37]. Secondly, as aging advances, the propensity to develop swallowing difficulties increases as a result of frequent neurologic damage or disorders, such as stroke [38], Alzheimer's disease [39], and Parkinson's disease [40]. Thus, there are substantial socio-economic implications related to age-related disease. Due to these reasons, it is crucial to recognize the impact of

aging on normal swallow so that its negative consequences on nutrition, hydration, pulmonary function, and overall quality of life can be prevented or addressed appropriately.

A few studies have addressed laterality aspects of swallowing in healthy older adults. Malandraki et al. [14] utilized functional magnetic resonance imaging (fMRI) and demonstrated such differences in healthy aging adults. The authors compared nine healthy aging adults to ten younger adults on voluntary swallowing of 3 ml of room temperature water, planning of a swallow without execution, tapping of the tip of the tongue against the alveolar ridge, and throat clearing tasks. The results of their study revealed a reduced lateralization for swallowing in the older adult group as compared to the younger adults. These findings are suggestive of a swallowing related hemispheric specialization preference in younger adults, and a potential reduction in hemispheric specialization for swallowing in older adults.

Martin et al. [41] also reported fMRI data that potentially suggest the recruitment of additional areas for swallowing in older adults. Although laterality was not directly addressed, the authors reported that the volume of brain areas involved in water swallow for older adults was substantially higher compared to younger adults suggesting that older adults tend to compensate for increased task demands by recruiting additional areas. Teismann et al. [15] reported similar results. These results are suggestive of a possible reduction/change in hemispheric specialization as a function of age.

Changes that occur to the aging brain with respect to hemispheric specialization of different cognitive and behavioral tasks have been explained by two predominant views, the right hemi- aging model [42], and the hemispheric asymmetry reduction in older adults (HAROLD) model [43]. The right hemi- aging model suggests that the right hemisphere shows more age-related decline, while the HAROLD model proposes that hemispheric specialization steadily decreases with age and that the brain recruits additional areas for functions that are specialized to be performed by one of the hemispheres in younger healthy adults.

## Theoretical constructs of the present study

Earlier studies [27–29] related to swallowing laterality have established the following constructs. 1) Swallowing is, to some extent, bilaterally controlled. 2) There is a differential hemispherical control for specific aspects of swallowing. That is, the left hemisphere demonstrates a preferential control for volume of swallow, while the right hemisphere is responsible for timing-related aspects. In continuation with earlier studies by Daniels et al. [28], and Balasubramanium et al [29], we aim to investigate potential reduction/changes in hemispheric specialization in healthy aging adults for swallowing related tasks using a behavioral dual-task paradigm.

We began by carefully selecting tasks, which would potentially draw neural resources for swallowing and a competing task. For the left hemisphere, a silent word repetition was considered as it has been reported that the neural regions activated for swallowing and motor speech are anatomically close and functionally overlapping [44,45]. Warburton et al., [46] have reported an activation of the left primary motor cortex during silent word repetition, and similar areas are reported to be involved during swallowing [2,3,9]. For the right hemisphere, facial recognition task was selected, as several imaging studies [47,48] have attributed the right fusiform face area in the occipitotemporal lobe to be responsible for visual facial recognition. Similar areas within the right hemisphere, especially the sensorimotor integration networks have been reported to contribute to swallowing [49]. These evidences from functional imaging studies support the premise that swallowing, motor speech, and visual facial recognition centres functionally overlap.

Based on what is already known in the areas of dual-tasking, hemispherical laterality/specialization for swallowing, and swallowing related neurological changes in healthy aging, the

following was hypothesized. 1) Performing concurrent cognitive and swallowing tasks (via a dual-task paradigm) would result in decrements to clinical swallowing performance in younger and older adults; with greater decrements in the former. This hypothesis was based on the notion that older adults experience constrained resource allocation while performing a dual-task. In other words, we hypothesized that younger adults would demonstrate a substantial attenuation in their performance of the swallowing task during interference conditions compared to baseline performance due to the interfering task demands. Older adults, because of the effects of healthy aging, may already demonstrate reduced specialization. Therefore, performance attenuation due to dual-task demands was hypothesized to be modest.

## Method

Human participants: The study was conducted after approval from the Research Ethics Committee at Kasturba Medical College, Mangalore, India.

Eighty healthy right-handed participants recruited from the community participated in the study. Participants were divided into two groups: Group 1 consisting of young adults (18–40 years) and Group 2 consisting of community-dwelling healthy aging adults (65 years and older). The handedness of the participants was confirmed using the Edinburgh Handedness Inventory [50]. Participants were recruited based on sample size calculations for a cross-sectional study, with an equal number of males and females. Exclusion criteria were a history of dysphagia, head, and neck structural damage, and neurological disorders. Inclusion criterion included a score of < 3 on the Kannada version of Eating Assessment Tool 10 (EAT 10 K) [51]. Cognitive dysfunction was ruled out using the Mini-Mental State Examination (MMSE) [52]. All participants had an MMSE score of 24 and above. Participants were included in the study after obtaining their written informed consent.

Each participant performed baseline and interference tasks. Baseline condition included continuous swallowing using the timed test of swallowing (TTS) [53]. The interference conditions with swallowing included a silent word repetition task for the left hemisphere [49] and a facial recognition task for the right hemisphere involvement [54,55]. For all the trials, the order of tasks was randomized and counterbalanced to prevent practice effect.

### Baseline swallowing task

Baseline swallow performance was assessed using two trials of TTS. Participants continually ingested 100 ml of water from a cup. They were instructed to swallow at a comfortable rate without spillage or pausing until asked to stop. Number of swallows (NS) during continuous cup drinking was measured using video recordings of the task and was averaged across two trials. Volume per swallow (VPS) was calculated by dividing the total amount ingested during each trial by the number of swallows and was averaged across two trials. Volume per time (VPT) was calculated by dividing total volume (100 ml) by total time and was averaged across two trials. Time per swallow (TPS) was calculated by dividing the total time taken to ingest 100 ml of water by NS and was averaged across two trials.

### Interference tasks

The interference tasks consisted of a right hemisphere task (facial recognition) and a left hemisphere task (silent word repetition). For the facial recognition task, participants were presented with images of eminent personalities that were available in public databases and were instructed to think of the answer but not vocalize it. The images were first presented to fifteen healthy individuals (who did not participate in the study) across the lifespan for item agreement. The images were validated for accuracy and clarity. Images were included only when

they were agreed by 100% of these individuals to be representative of the personality concerned and were of high resolution. Each of the stimuli was presented using Microsoft Power-Point for a total of three seconds.

For the left hemisphere task, participants silently repeated stimuli from a word set namely, wolf, butterfly, and duck [27,28] as they continuously swallowed. Participants did not vocalize the word set during this task; rather they were instructed to rapidly and repeatedly think of the stimuli in word set. After each trial, confirmation of silent repetition was obtained. For each of the baseline and competing tasks, the swallow performance was measured on VPS, VPT, and TPS.

### Statistical analysis

Mixed model Analysis of variance with appropriate Bonferroni corrections was performed, with the group (young vs. older adults) as the between-subject variable and interference tasks as the within-subject variable. The independent variables (IV) were age and interference conditions; dependent variables (DV) were VPS, VPT, and TPS.

## Results

Descriptive data for swallowing performance of two groups with and without interference conditions are presented in Figs 1–3. A single mixed model analysis of variance, with appropriate Bonferroni corrections was performed with the group (young vs. older adults) as the between-subject variable, and interference tasks as the within-subject variable. Results are discussed below under the following headings.

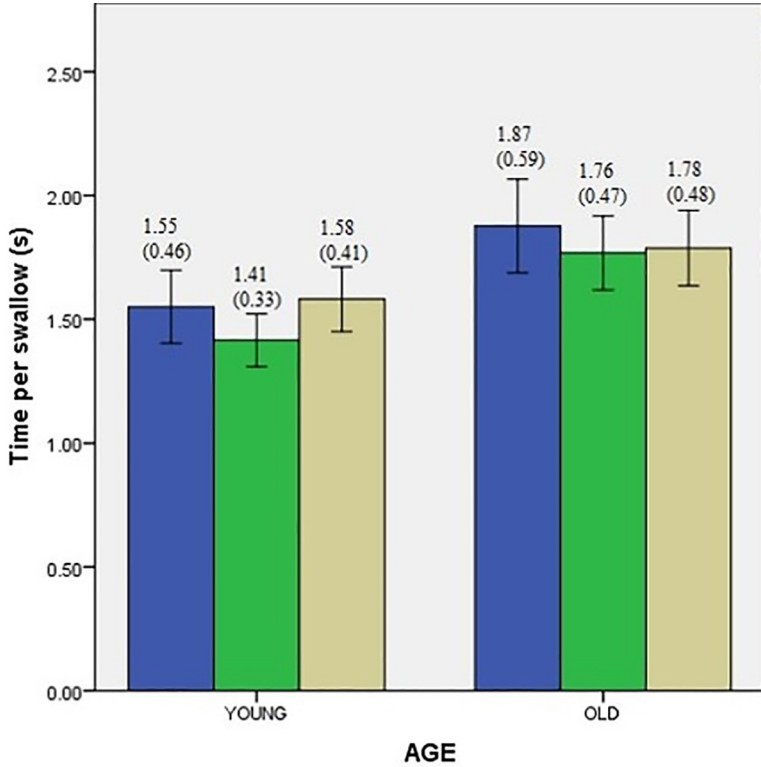

**Fig 1. Volume per swallow comparisons of younger and older adults during conditions of interference represented as mean (standard deviation).** Error bars indicate standard error (SE).

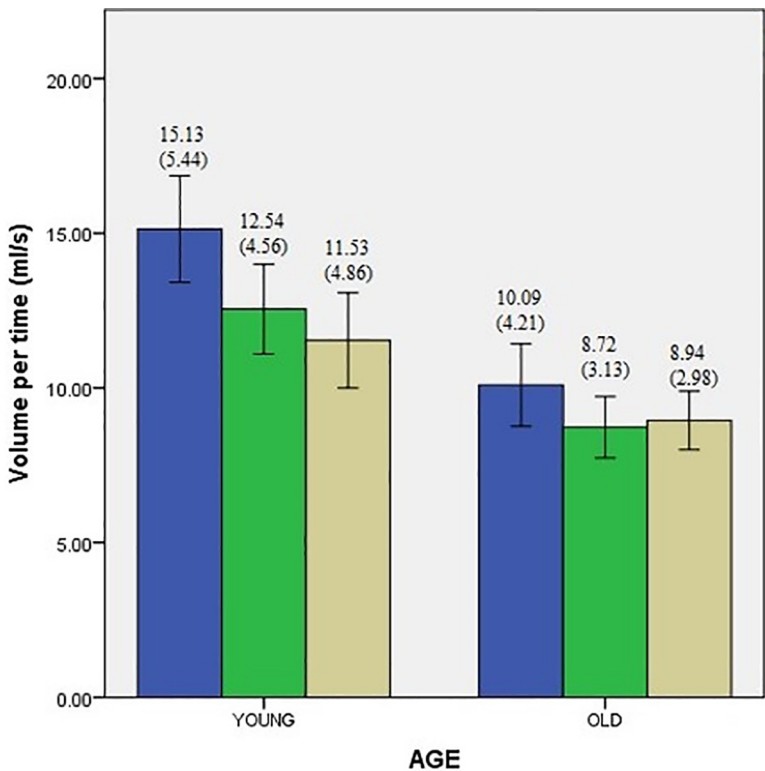

**Fig 2. Volume per time comparisons of younger and older adults during conditions of interference represented as mean (standard deviation).** Error bars indicate standard error (SE). * Indicates statistical significance.

### Effect of age on swallow performance

Means for swallow performance between the age groups are presented in Figs 1–3.

To investigate the overall effects of aging on swallowing performance, a mixed model ANOVA with group (young vs. older adults) as between subject factor was performed. Results demonstrate a significant effect of age for all three measures of swallow performance. The measures of VPS ($F_{(1,78)}$ = 5.362, $p$ = 0.05 $d$ = 0.29), and VPT ($F_{(1,78)}$ = 17.34, $p$ = 0.05 $d$ = 1.03) were found to be significantly lower in older adults compared to young adults. The measure of TPS ($F_{(1,78)}$ = 4.097 $p$ = 0.05 $d$ = 0.6) was significantly higher in older adults compared to younger adults.

### Effect of interference conditions on swallow performance

Means for swallow performance during each of the interference conditions is represented in Figs 1–3. To investigate how the two interference conditions (facial recognition and silent word repetition) influenced the swallowing performance (measured in terms of VPS, VPT, and TPS) in the two groups (young vs. older adults), mixed model ANOVA was performed with the group as the between-subject variable and interference tasks as the within-subject variable. The results are presented with respect to each dependent variable (VPS, VPT, and TPS) below.

**Volume per swallow.** The results demonstrate a significant main effect of interference ($F_{(1.572,122)}$ = 6.198, $p$ = 0.03). However, there was no significant interaction between age and the interference tasks ($F_{(1.577,122)}$ = 1.508, $p$ = 0.05). Bonferroni post hoc pairwise comparisons further demonstrated a statistically significant decrease in both the interference conditions baseline–left hemisphere task (silent word repetition) ($p$ = 0.01), and baseline–right hemisphere task (facial recognition) ($p$ = 0.01).

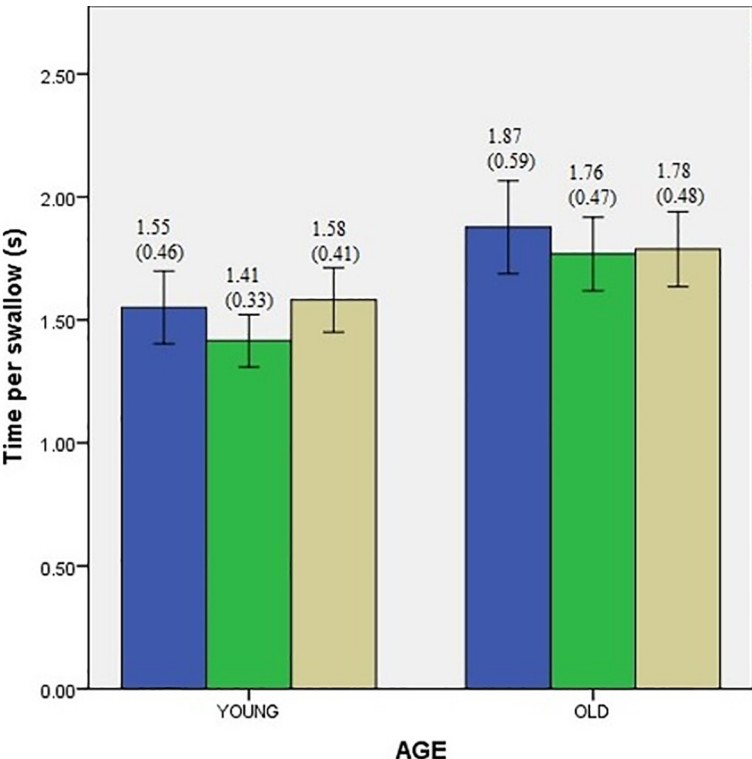

**Fig 3. Time per swallow comparisons of younger and older adults during conditions of interference represented as mean (standard deviation).** Error bars indicate standard error (SE).

**Volume per time.** Results demonstrate a statistically significant main effect of interference ($F_{(2.84,221.5)}$ = 15.330, $p$ = 0.05), and also an interaction between age and the interference conditions ($F_{(2.81,221)}$ = 3.646, $p$ = 0.05). Bonferroni post hoc pairwise comparisons further demonstrated a statistically significant decrease in baseline–left hemisphere task (silent word repetition) ($p$ = 0.01), and baseline–right hemisphere task (facial recognition) ($p$ = 0.01).

Since there was a significant interaction between age and the interference condition, repeated measures ANOVA was performed separately for young adult and older adult groups. For the young adults group, one-way repeated measures of ANOVA was performed with the interference tasks as the within group variable. A significant main effect of the interference tasks ($F_{(3.087,120.3)}$ = 16.544, $p$ = 0.05) was observed. Bonferroni post hoc pairwise comparisons showed a statistically significant decrease in baseline–left hemisphere task (silent word repetition) ($p$ = 0.01), and baseline–right hemisphere task (facial recognition) ($p$ = 0.01).

For the older adults group, the results of one-way repeated measures ANOVA revealed a statistically significant main effect of the interference ($F_{(2.443,94.894)}$ = 3.844, $p$ = 0.05). Bonferroni post hoc pairwise comparisons did not reveal any significance between baseline–left hemisphere task (silent word repetition) ($p$ = 0.92), or baseline–right hemisphere task (facial recognition) ($p$ = 0.9) conditions.

**Time per swallow.** Results did not reveal a significant main effect of the interference ($F_{(1.598,124.6)}$ = 1.01, $p$ = 0.09), nor any interaction between age and interference ($F_{(1.598,124.6)}$ = 1.931, $p$ = 0.13) was found. In summary, the interference of silent word repetition (left hemisphere task) yielded a statistically significant effect on volume per swallow and volume per time for the younger adults and only on volume per swallow for the older adults. No

statistically significant differences were observed for time per swallow for either of the groups and volume per time for the older adults group. The interference of facial recognition (right hemisphere task) demonstrated a similar pattern as the left hemisphere interference. That is, there was a statistically significant decrement in volume per swallow and volume per time for the younger adults and only volume per swallow for the older adults. In other words, younger adults swallowed lesser volumes of boluses (VPS and VPT) during the interference conditions compared to baseline. While statistical significance was not observed for the TPS parameter, performance tended to deteriorate in the interference conditions characterized by the increased time to swallow for younger adults, as can be seen in Fig 3. In addition, older adults did not show a statistically significant effect of the conditions of interference on two of the three swallowing parameters that we investigated.

## Discussion

The study investigated potential changes in hemispheric laterality with respect to swallowing in healthy aging adults. The results are suggestive of 1) a statistically significant effect of age on swallow performance; 2) statistically significant effect of each of the interference tasks on two of the swallow measures (VPS and VPT) in younger adults and trends toward decrement for the TPS parameter, and; 3) No significant effect of either of the interference tasks on two parameters (VPT and TPS) for older adults.

The first key finding of an effect of age on swallowing performance was on expected lines. Aging substantially impacts both neurological functions and peripheral musculature interfering with the ability to swallow safely and efficiently [56–61] and the results from our data are consistent with what is already known. The second key finding of an effect of the interference tasks on swallowing performance also makes sense and is consistent with our hypothesis. Competing tasks that utilize similar neurological structures require differential allocation of neural resources. In the present study, silent word repetition and facial recognition were adopted as interfering tasks and these tasks utilize areas in the left and right hemispheres respectively, that overlap with swallowing related functions. Whereas our understanding of laterality related to swallowing continues to grow, several accounts report preferential laterality for specific components related to swallow (e.g. Malandraki and colleagues [14]). So, it does make clinical sense to observe the impact of interference tasks on swallowing measures. Interestingly, this impact of the interference tasks on swallowing performance was significantly evident in younger adults compared to older adults and is partly consistent with our hypothesis. Specifically, the parameter–VPS–did not demonstrate statistically significant interference effects in the older adult group. One possible explanation for this finding is that the dual-task paradigm is perhaps not sensitive to address volume-related aspects of swallowing. The results of the interference tasks should also be interpreted with caution because both the interference tasks did not require an overt response.

The findings of the study appear to shed light on the patterns of laterality changes in older patients. Regardless of conditions that are known to interfere with swallowing performance warranting division of neural resources [28], the performance of older adults on two of the three swallowing related parameters we investigated, tended to be rather similar to that when the interference conditions were not present. While it is premature to attribute all these findings to a reduced asymmetry in aging, it does appear that one possible reason for the differences in performance could be age-related changes in the brain. This study did not measure neural activation, however; the findings are supported by earlier reports on the HAROLD model on aging that suggests that the activity of the prefrontal cortex tends to be less lateralized in older than in younger adults [46]. Data from behavioral experiments investigating age-

related neuropsychological and cognitive functions of memory, perception, inhibition, etc. have demonstrated a reduction in hemispheric specialization in older adults [46] and our data are consistent with these findings. With respect to swallowing, studies that have utilized functional imaging to investigate neural control of swallowing [14,41] have all suggested changes in brain activation and potentially increased activation suggesting compensatory utilization of neural resources.

Making over-reaching conclusions based on behavioral data is rash. Nevertheless, the findings add support to the growing evidence for a possible reduction in the specialization of hemispheres in healthy aging and yet demonstrating behavioral performance at below optimum levels. This reduction in hemispheric specialization could be due to several reasons including a compensatory mechanism where the reduced asymmetries could help with the deficits that occur with aging [62] or a de-differentiation mechanism which suggests that reduction in asymmetries are simple by-products of aging [63]. Other proposals such as a change in the cognitive architecture [64] or the general neural networks [65] with aging have found evidence as well. Considering all the findings together, older individuals appear to demonstrate poorer swallowing outcomes in spite of a possible change in their hemispheric laterality. In other words, the reduction in the asymmetries does not appear to offer a compensatory preservation of swallowing measures in healthy aging individuals.

## Conclusions

In conclusion, aging substantially impacts swallowing in older individuals and this potentially accompanies a reduction in the hemispheric specialization for swallowing-related tasks among other possible changes. Future research should objectively substantiate these possible age related hemispheric changes and investigate if these changes help reduce the negative impacts of aging on the swallowing musculature and function combining multimodal outcomes.

## Supporting information

**S1 Dataset. Swallow laterality data sheet.**
(XLSX)

## Author Contributions

**Conceptualization:** Rahul Krishnamurthy, Radish Kumar Balasubramanium, Balaji Rangarathnam.

**Formal analysis:** Rahul Krishnamurthy, Radish Kumar Balasubramanium, Balaji Rangarathnam.

**Methodology:** Rhea Philip, Balaji Rangarathnam.

**Project administration:** Rahul Krishnamurthy.

**Supervision:** Radish Kumar Balasubramanium.

**Writing – original draft:** Rahul Krishnamurthy.

**Writing – review & editing:** Radish Kumar Balasubramanium, Balaji Rangarathnam.

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
