## [Decision Letter · Decision Letter 0]

11 May 2021

PONE-D-21-01457

Effects of dual-task interference on swallowing in healthy aging adults

PLOS ONE

Dear Dr. Rangarathnam,

Thank you for submitting your manuscript to PLOS ONE. After careful consideration, we feel that it has merit but does not fully meet PLOS ONE’s publication criteria as it currently stands. Therefore, we invite you to submit a revised version of the manuscript that addresses the points raised during the review process.

In particular, one referee felt that some details of the statistical analysis and potential limitations of the interpretations should be better specified. 

We look forward to receiving your revised manuscript.

Kind regards,

Giovanni Petri, Ph.D.

Academic Editor

PLOS ONE

Journal Requirements:

3. In line with PLOS' guidelines on detailed reporting (https://journals.plos.org/plosone/s/criteria-for-publication#loc-3), please ensure that you have provided sufficient detail in the Methods section on how and from where younger adult participants were recruited.

4. Please improve statistical reporting and report exact p-values for all values greater than or equal to 0.001. Our statistical reporting guidelines are available at https://journals.plos.org/plosone/s/submission-guidelines#loc-statistical-reporting.

Reviewers' comments:

Reviewer's Responses to Questions

**Comments to the Author**

1. Is the manuscript technically sound, and do the data support the conclusions?

Reviewer #1: Yes

Reviewer #2: Partly

2. Has the statistical analysis been performed appropriately and rigorously? 

Reviewer #1: Yes

Reviewer #2: Yes

3. Have the authors made all data underlying the findings in their manuscript fully available?

Reviewer #1: Yes

Reviewer #2: Yes

4. Is the manuscript presented in an intelligible fashion and written in standard English?

Reviewer #1: Yes

Reviewer #2: Yes

5. Review Comments to the Author

Reviewer #1: The study was aimed at investigating differences between young and old participants in their swallowing performance in basal conditions and during tasks engaging the left or right hemisphere. Results indicate that old age suppresses laterality differences and support the view that the left and right hemishere are involved in different parameters of swallowing.

The study is interesting and well conducted. The manuscript is well written.

Findings are relevant to basic and clinical science

Reviewer #2: The authors present a novel and well-written study into possible differences in hemispheric dominance and swallowing between groups of younger and older adults. Younger adults showed decrements to volume per time swallowing behaviours when performing silent word recognition (assumed left hemisphere dominant task) and facial recognition (assumed right hemisphere dominant task). Although older adults showed general decrements on all measures relative to younger adults, no dual-tasking related interference is observed.

I think that the study is well powered and the methodological detail is excellent. I just have the following comments:

Interpretation

The authors do well to clarify how it is important to not overreach conclusions regarding hemispheric dominance based on behavioural data alone. I think that the following interpretations should at least be considered in the discussion:

i) is it certain that the finding that older adults did not show a significant interaction on the volume per time measure of swallowing behaviour due to a floor effect? The VPT for older adults looks substantially lower than that for younger adults, and so I wonder what the range on this measure actually is - i.e. is the measure sensitive/appropriate to pick up interference in older adults?

ii) the interference task did not require an overt response. Therefore, could it be that older adults expressed greater performance decrements on the task for which there are no behavioural measurements?

Results

The results are clear but some critical details are missing that would inform interpretation and enable future meta-analyses - specifically, could the authors present attained p-values, effect sizes, and confidence intervals on those effect sizes?

Re: the figures - it is unclear what the error bars are depicting. Could the authors amend the figure captions to make this clear?

Figures 1-3 seem unnecessary, as the data are presented again in Figures 5-6.

Introduction

From the introduction on hemispheric specialization on swallowing, I was unclear exactly what the fMRI and PET data suggest re: hemispheric dominance - i.e. is it the left or the right hemisphere? Is the literature convergent in the findings?

The Daniels et al [27] paper seems particularly important for the study. It would be good to be able to understand how those specific conclusions were reached re: left vs right dominance for volume and temporal parameters of swallowing.

Re: the Malandraki et al [14] study - it is not clear what was found for the older adults and thus what could be concluded. I am guessing that there were no hemispheric differences in older adults, but was there a statistically significant age-group x laterality interaction?

6. PLOS authors have the option to publish the peer review history of their article (what does this mean?). If published, this will include your full peer review and any attached files.

Reviewer #1: **Yes: **ENRICA L. SANTARCANGELO

Reviewer #2: **Yes: **Kelly Grace Garner

---

## [Author Response · Author response to Decision Letter 0]

1 Jun 2021

May 24, 2021

Dear Dr. Petri:

I am submitting a revised version of the manuscript PONE-D-21-01457: Effects of dual-task interference on swallowing in healthy aging adults. I would like to thank you and the reviewers: Drs. Santarcangelo and Garner for the thoughtful review of the manuscript and invaluable suggestions to improve it. We have incorporated the suggested changes into the manuscript. Please see our responses to each suggestion below:

Editorial comments

Sl. No Editor comments Action taken

1. Please ensure that your manuscript meets PLOS ONE's style requirements, including those for file naming. The manuscript has been revised according to PLOS ONE's style requirements.

2. Please provide additional details regarding participant consent. In the ethics statement in the Methods and online submission information, please ensure that you have specified (1) whether consent was informed and (2) what type you obtained (for instance, written or verbal, and if verbal, how it was documented and witnessed). If your study included minors, state whether you obtained consent from parents or guardians. If the need for consent was waived by the ethics committee, please include this information. Has been reported in the methods section.

Page 7; Lines - 139 – 141.

3. In line with PLOS' guidelines on detailed reporting (https://journals.plos.org/plosone/s/criteria-for-publication#loc-3), please ensure that you have provided sufficient detail in the Methods section on how and from where younger adult participants were recruited.

 Has been reported in the methods section. Page 7; Line 130.

4. Please improve statistical reporting and report exact p-values for all values greater than or equal to 0.001. 

Exact p values have been reported.

5. We note that you have indicated that data from this study are available upon request. PLOS only allows data to be available upon request if there are legal or ethical restrictions on sharing data publicly. 

Yes, the data spreadsheet has been added as a "supporting document".

Reviewer Comments:

 Interpretation

The authors do well to clarify how it is important to not overreach conclusions regarding hemispheric dominance based on behavioural data alone. I think that the following interpretations should at least be considered in the discussion:

i) is it certain that the finding that older adults did not show a significant interaction on the volume per time measure of swallowing behaviour due to a floor effect? The VPT for older adults looks substantially lower than that for younger adults, and so I wonder what the range on this measure actually is - i.e. is the measure sensitive/appropriate to pick up interference in older adults?

We thank the reviewer for this observation. However, we respectfully disagree on the possibility of a floor effect. The values for VPT in the current study are well within the normative range for healthy older adults.

Mean (SD): 14.68 (6.11).

We have acknowledged that volume related measures (VPT, and VPS) may be immune to dual tasking effects. 

Pg. No – 14; Lines 282-284

 the interference task did not require an overt response. Therefore, could it be that older adults expressed greater performance decrements on the task for which there are no behavioural measurements? 

We have added this caveat in the discussion section.

 Results

The results are clear but some critical details are missing that would inform interpretation and enable future meta-analyses - specifically, could the authors present attained p-values, effect sizes, and confidence intervals on those effect sizes? 

Exact p values have been reported.

Effect sizes: Cohen’s d has been reported for comparisons with statistical significance.

Re: the figures - it is unclear what the error bars are depicting. Could the authors amend the figure captions to make this clear?

Figures 1-3 seem unnecessary, as the data are presented again in Figures 5-6. Figures 1 – 3 have been removed.

Description for the figures have been provided.

 Introduction

From the introduction on hemispheric specialization on swallowing, I was unclear exactly what the fMRI and PET data suggest re: hemispheric dominance - i.e. is it the left or the right hemisphere? Is the literature convergent in the findings?

Changes have been made to the statement for better interpretation. Pg. No – 2; Lines 35-38

The Daniels et al [27] paper seems particularly important for the study. It would be good to be able to understand how those specific conclusions were reached re: left vs right dominance for volume and temporal parameters of swallowing.

 Changes have been made to the statement. Pg. No – 3; Lines 61-64

Re: the Malandraki et al [14] study - it is not clear what was found for the older adults and thus what could be concluded. 

I am guessing that there were no hemispheric differences in older adults, but was there a statistically significant age-group x laterality interaction? 

This statement is revised for better clarity. Pg. No – 4; Lines 84-87

Thanks again for your conscientious review with this manuscript. We look forward to hearing more about the review of this resubmission.

Sincerely,

Balaji Rangarathnam

---

## [Editor Report · Decision Letter 1]

8 Jun 2021

Effects of dual-task interference on swallowing in healthy aging adults

PONE-D-21-01457R1

Dear Dr. Rangarathnam,

We’re pleased to inform you that your manuscript has been judged scientifically suitable for publication and will be formally accepted for publication once it meets all outstanding technical requirements.

Kind regards,

Giovanni Petri, Ph.D.

Academic Editor

PLOS ONE
---

## [Editor Report · Acceptance letter]

15 Jun 2021

PONE-D-21-01457R1 

Effects of dual-task interference on swallowing in healthy aging adults 

Dear Dr. Rangarathnam:

I'm pleased to inform you that your manuscript has been deemed suitable for publication in PLOS ONE. Congratulations! Your manuscript is now with our production department. 

Kind regards, 

on behalf of

Dr. Giovanni Petri 

Academic Editor

PLOS ONE